# Specific Radiologic Risk Factors for Implant Failure and Osteonecrosis of the Humeral Head after Interlocking Nailing with the Targon PH^+^ of Proximal Humeral Fractures in a Middle to Old Population

**DOI:** 10.3390/jcm11092523

**Published:** 2022-04-30

**Authors:** Maximilian Willauschus, Linus Schram, Michael Millrose, Johannes Rüther, Kim Loose, Hermann Josef Bail, Markus Geßlein

**Affiliations:** 1Department of Orthopedics and Traumatology, Paracelsus Private Medical University Nuremberg, 90419 Nuremberg, Germany; linus.schram@stud.pmu.ac.at (L.S.); johannes.ruether@klinikum-nuernberg.de (J.R.); kim.loose@klinikum-nuernberg.de (K.L.); hermann-josef.bail@klinikum-nuernberg.de (H.J.B.); markus.gesslein@klinikum-nuernberg.de (M.G.); 2Department of Trauma Surgery and Sports Medicine, Garmisch-Partenkirchen Medical Centre, 82467 Garmisch-Partenkirchen, Germany; michael.millrose@klinikum-gap.de

**Keywords:** proximal humeral fractures, humeral head necrosis, interlocking antegrade nailing, complications, implant failure, risk factors

## Abstract

Background: Operative management of proximal humeral fractures is still challenging. While antegrade nailing has become a valid option in fracture fixation, risk factors for adverse events, and failure have not been sufficiently clarified. Methods: All patients of a single trauma center undergoing surgery for proximal humeral fractures with the Targon PH^+^ nail between 2014 and 2021 were evaluated retrospectively. This included complications, revisions, and failures. Pre- and postoperative radiographic imaging were assessed regarding fracture’s complexity, anatomic reduction, reconstruction of the medial hinge, metaphyseal head extension, and fixation of the implant in the calcar region. Follow-up was at a minimum of 12 months. Results: A total number of 130 patients with a mean age of 74.5 years (range 63–94, SD ± 8.2) are included in this study. Two- and three-part fractures were found in 58 patients, while 14 patients showed four-part fractures. Overall, a complication rate of 34.2% and an implant failure rate of 15.4% was found. Four-part fractures showed a significantly higher complication rate than two- and three-part fractures. Four-part fractures also showed significantly higher revisions (*p* = 0.005) and implant failures (*p* = 0.008). The nonsufficient anatomical reduction was found to be a risk factor for complications (*p* < 0.0001), implant failures (*p* < 0.0001), and later humeral head osteonecrosis (*p* < 0.0001). Insufficiently reconstructed medial hinges (*p* = 0.002) and a metaphyseal head extension of over 8 mm (*p* = 0.005) were also demonstrated as risk factors for osteonecrosis of the humeral head. Conclusions: Four-part fractures in an elderly population show high complication, revision, and implant-failure rates. Therefore, demonstrated radiologic risk factors should be evaluated for improvements. Anatomical reduction and fixation near the calcar proved to be vital for successful antegrade nailing of complex fractures. To prevent osteonecrosis of the humeral head, reconstruction of the medial hinge and metaphyseal head extension should be evaluated.

## 1. Introduction

Proximal humeral fractures are the third most common fracture in elderly patients over 65 years. Non-displaced stable fractures can be treated conservatively [1,2,3]. Displaced and unstable fractures, including two-, three-, and four-part, should be treated operatively. Plate fixation, intramedullary nailing, percutaneous bridging systems, and hemiarthroplasty are the commonly applied treatments [4,5]. The aims during surgery are to restore anatomy and preserve blood supply to the humeral head and sustain the shoulder joint’s physiologic function [6,7,8]. There is still debate concerning the superiority of any fixation method [9]. The choice of the method primarily depends on a patient’s age, functional needs, bone quality, fracture pattern, and the surgeon’s expertise.

Antegrade intramedullary interlocking nailing seems to be a safe and efficient method in the hands of an experienced surgeon [10,11]. Particularly in the case of elderly patients, it enables a minimally invasive approach reducing additional trauma and potential complications. The Targon PH^+^ (B. Braun Aesculap, Tuttlingen, Germany) was specially intended to provide a locking head fixation for several humeral fractures. The proximal locking head screws are ascendingly designed to increase the hold in the osteoporotic bone while maintaining the blood supply [12].

Multiple complications can occur and have been widely reported in the related literature [13]. Complication rates can be high, and risk factors for failure of interlocking nailing have not yet been satisfactorily clarified.

This study aimed to evaluate specific risk factors, especially, adverse events and failure of antegrade nailing of the proximal humerus in an elderly population while examining specific radiologic risk factors for osteonecrosis of the humeral head.

## 2. Materials and Methods

All patients (*n* = 251) with proximal humeral fracture treated with antegrade interlocking nailing in the period from 2014–2021 at a level-1 trauma center were evaluated in this retrospective analysis.

The study was approved by the institutional review committee (Paracelsus Medical University at the Nuremberg Hospital, Number IRB-2021-025) following national legal guidelines. All procedures that were executed complied with the ethical principles of the institutional, national research committee, and the 1975 Helsinki declaration. Informed consent was obtained from all patients participating in the study.

Inclusion criteria were unilateral proximal fracture of the humerus, age over 60, follow-up radiographs after at least 12 months, and fracture fixation with antegrade interlocking nailing using the Targon PH^+^. 

Exclusion criteria were death (during the hospital stay or during the follow-up period), pathological fractures due to tumorous destruction, combined shaft fractures, long versions of the nail (>150 mm), open fractures, and patients requiring initial shoulder arthroplasty. Patients with missing radiographs and those who could not give their consent to the study have also been excluded.

Based on these criteria, 130 patients were included in this study.

The decision for nailing was decided by a board of experienced orthopedic and traumatological experts.

All data regarding patient history, pre-existing conditions, complications, concomitant fracture or injuries, lab work, implant failure, and re-operations were taken from the patient’s hospital records.

The pre- and postoperative radiographic imaging was evaluated on the (PACS) software using plain radiographic images or CT scans. All fractures were classified according to the degree of fractured parts including the Neer classification [14]. According to the Neer classification, fractures with single involvement of the surgical neck were classified as III, 2- and 3-part fractures with involvement of the greater tuberosity as IV, of the lesser tuberosity as V, and with luxation of the head as VI. Four-part fractures with involvement of both the lesser and greater tuberosity and the surgical neck were classified as IV/4-part [14]. Post-op anatomic reposition, the integrity of the medial hinge of the proximal humerus, and the medial metaphyseal extension length were evaluated, as also the pre- and postoperative neck-shaft angle and the tip of the inferior head screw to calcar distance.

The local bone quality of the proximal humerus was assessed via the calculation of the deltoid tuberosity index (DTI), the medial cortical ratio (MCR), and the Tingart measurement (TM) in the preoperative radiographic imaging.

Follow-up radiographs were reviewed for recorded complications by two experienced trauma surgeons (Figure 1 and Figure 2). An anatomic reposition was suspected when the head was neither displaced in varus or valgus (<20° of the normal neck shaft angle 135°), no anterior or posterior tilt of the head greater than 20° on the y-view was present, dislocation of the greater or lesser tuberosity was less than 3 mm, and a neck-shaft dislocation of less than 5 mm was obtained.

Adverse events associated with the procedure, including implant and patient-related complications and treatment failures, were also taken directly from patients’ records. Patients were only included if they could be evaluated after 12-month follow-up period including a radiographic control after 12 months or longer.

### 2.1. Surgical Technique

Prior to surgery, patients were positioned in a beach chair position on a radiolucent table with a standard armrest. For implantation of the Targon PH^+^ nail, a deltoid splitting approach at the anterolateral margin of the acromion was performed. After a longitudinal transection of the clavipectoral fascia and the subacromial bursa, the head fragments were reduced by indirect manipulation. For this procedure, a wire was used as a joystick for securing the fragments. After the longitudinal splitting of the supraspinatus tendon, a guide pin at the apex of the humeral head was positioned in both planes, and the nail was inserted 3 to 4 mm below the cartilage level. Then, a minimum of three to four locking head fixation screws were inserted into the head fragments, depending on the fracture pattern, the degree of instability, as well as bone quality. To secure displaced fragments to the interlocking screws, a “rope-over-bitt” procedure was carried out in cases of highly displaced tubercula (>1 cm).

A single or duplicate distal interlocking screw was placed through the nail percutaneously. Board-certified trauma surgeons performed all operations. A postoperative x-ray examination was obtained from all patients to verify both repositioning and correct positioning of the implant.

The affected arm was immobilized in a sling bandage for one day to control postoperative pain. On the second postoperative day, a passive mobilization by trained physiotherapists was started. An active-assisted shoulder movement below the pain threshold was then initiated as soon as wound healing was deemed satisfactory. After six weeks and radiological consolidation, weight-bearing was allowed.

Every patient received postoperative standardized anterior-posterior and y-view radiographic imaging for correct fracture reposition and screw position after passive mobilization. A radiographic follow-up after 12 months or longer was obtained from all patients.

### 2.2. Evaluation of Complications

Early postoperative complications were evaluated during the hospital stay of the patients. These included reviews of post-operative X-rays and medical records. Later, complications were recorded from hospital charts during visits at our outpatient clinic or if the patient was re-admitted to the hospital. General complications were recorded as well as implant-specific complications.

### 2.3. Statistical Analysis

All data were obtained and analyzed retrospectively. Statistical analysis was performed using IBM SPSS Statistics for Windows (version 28, 1.0.0.1406, IBM Corp., Armonk, NY, USA).

Parametric data (DTI, TM, and MCR) were analyzed using *t*-test. For nominal data (anatomic reduction, complications, revision, implant failure, etc.), the Pearson chi-square test or Fisher’s exact test were used with the corresponding *p*-value. The Fisher exact test usage cutoff value was set at *n* < 10 in one or more categories. The Pearson correlation coefficient was calculated for correlations (DTI, TM, and MCR). Assessed risks were analyzed, calculating the relative risk (RR) and odds ratio (OR) out of cross tables. If OR and RR could not be calculated, the risk difference (RD) was utilized. All reported *p*-values are two-tailed, with an alpha level < 0.05 considered statistically significant. Unless otherwise stated, descriptive results are demonstrated as mean ± standard deviation and range.

## 3. Results

One hundred thirty patients met the inclusion criteria and were evaluated. The average age was 75.5 years (range 63–94, SD ± 8.2). Gender distribution was 65.2% female and 35.8%, male. A total of 115 (88.5%) patients had pre-existing chronic conditions and 49 (37.6%) were classified as multimorbid (>3 chronic illnesses). More specifically 37 (28.4%) patients had a pre-existing neurologic condition, 32 (24.6%) a cardiac disease, 32 (24.6%) a pulmonary disease. A total of 10 (7.6%) patients abused nicotine (daily use), and 17 (13.0%) alcohol (>20 g per day),while 4 (3.1%) abused other drugs.

Two-part fractures accounted for 44.6%, three-part fractures for 44.6%, and 10.9% for four-part fractures. The average radiographic follow-up period was 17.5 months (range 12–47, SD ± 7.3). Characteristics of patients are demonstrated in Table 1.

The mean DTI and MCR in the study population were both under the cut-off levels for low local bone quality of 1.4 and 0.16 respectively with 1.39 (range: 1.18–1.86, SD ± 0.11) and 0.15 (range: 0.08–0.25, SD ± 0.031), respectively. The mean TM was found above the cut-off value for osteoporosis of 6 with 6.23 (range: 3.12–13.55, SD ± 1.56). There was no difference found in DTI, MCR, and TM regarding fracture classification and the number of fractured parts.

Therefore, 44 patients (33.8%) were diagnosed with osteoporosis at the time of the fracture. A significant association between the Tingart measurement and the presence of osteoporosis (*p* = 0.023) was found with a corresponding odds ratio (OR) of 2.37 and relative risk (RR) of 1.83.

Adverse events during the follow-up period were found in 45 cases (34.6%) (Table 2 and Table 3). A total of 39 (88.6%) of them were directly related to the implant (Table 2). A total of 34 (26.1%) revisions were performed, while 24 (15.4%) had complications leading to implant failure with subsequent removal of the nail.

Twenty (13.8%) had implant-specific complications that led to implant failure. Ten (7.6%) patients received total shoulder arthroplasty during revision surgery. In the entire study population, osteonecrosis of the humeral head (OHN) was the most common implant-related complication being mostly prevalent in four-part fractures (*n* = 3 out of 14, incidence = 21.4%). In two-part and three-part fractures, the OHN also showed the highest prevalence with 3 patients out of 58 patients each resulting only in an incidence of 5.2% each.

Assessing the integrity of the medial hinge of proximal humerus showed a preoperative disruption (>2 mm) in 109 cases (83.2%). In the case of four-part fractures, all 14 (100%) showed a preoperative disrupted medial hinge compared to 49 (84.5%) in the two-part and 45 (77.6%) in the three-part group. Through the operative measures, a reconstruction was achieved in 73 cases (66.9%). Out of the four-part group, in 9 (64.3%) patients the medial hinge could be reconstructed in comparison to 42 (72.4%) in the two-part and 43 (74.1%) in the three-part group. The mean metaphyseal head extension (MHE, also known as calcar extension) was 17.9 mm (range: 5.0–57.9 mm, SD ± 11.5 mm). A total of 71 (54.2%) patients had an MHE of under 8 mm as a proposed cut-off value for high risk of OHN by Hertel et al. [15].

A postoperative anatomic repositioning of the fracture could be achieved in 66 patients (50.7%). In 75 cases (57.7%), an anchorage near the calcar of the nail through the distal head screw (distance to calcar < 12 mm [16]) was noted.

When comparing four-part fractures to two- and three-part fractures, they showed a higher complication rate with significantly more revision procedures (*p* = 0.005) and a significantly higher implant failure rate (*p* = 0.008). All the complications in four-part fractures led to a revision and in 77.6% of cases to failure of the implant (Table 4).

The anatomical reduction of proximal humeral fractures seemed to be a protective factor for adverse events. When dividing the patient collective into two subgroups (anatomical reduction [AR] vs. no anatomical reduction [NAR]) there was a strong significant difference in complications (Table 5).

Moreover, the success rate of anatomical reduction was smaller in four-part fractures (28.6%) compared to two- (58.6%) and three-part (50.0%) fractures.

These findings were underscored by a greater occurrence in loss of stability (incl. implant loosening, cut out, loss of reposition, and cranial nail migration) of the fracture fixation in the cases of non-anatomical reposition (*p* < 0.0001, OR = 9.0, RR = 6.5). The non-anatomical reduction was also shown to be a risk factor for the development of later humeral head necrosis (*p* = 0.016/OR = 14.0/RR = 8.9).

Considering humeral head necrosis, the preoperative integrity of the medial hinge did not significantly influence the odds for an OHN. However, every patient with a disrupted preoperative medial hinge developed OHN (*p* = 0.355/see Table 6). If not restored, a significantly higher risk for an OHN was recorded (*p* = 0.002/see Table 6).

If a metaphyseal head extension less than 8 mm was measured, a significantly higher risk for an OHN was observed (*p* = 0.038/OR = 7.5/RR = 6.7). Six out of 50 (12%) patients with no anchorage of the nail close to the calcar developed a cut out versus none from the 75 patients with an anchorage leading to a significant difference (*p* = 0.005).

The measurements of local bone quality did not show any significant predictive value regarding all measured outcome parameters.

## 4. Discussion

The most important finding of this study is the higher complication rate in four-part fractures compared to two- and three-part fractures (31% in two- and three-part fractures vs. 64% in four-part fractures). While two- and three-part fractures demonstrated a low complication rate even in an elderly population, nailing of four-part fractures showed significantly higher complication and revision rates, and an increasing number of implant failures.

Greenberg et al. [17] compared nearly thirty patients with three-part and four-part fractures, which were treated with the same interlocking nail (Targon PH, B. Braun Aesculap, Tuttlingen, Germany). Three-part fractures showed better clinical outcome parameters, but no difference in failure rates was found.

Kloub et al. [18] retrospectively analyzed four-part fractures treated with antegrade nailing. In this cohort, the complication rate was comparable (57%) with a lower implant failure rate (20% vs. 47% in this study).

Lange et al. [19] showed similar complication rates using the same implant in a comparable number of patients, recording an overall complication rate of 37% and a revision rate of 32%. Contrary to our findings, the complication rate was the highest in the three-part group.

A review conducted by Congia et al. [20], including mainly retrospective studies, reported revision rates ranging from 0–30%, resulting in a mean rate of 17%. It is noted that in this review, most patients included had two- and three-part fractures.

In an older study conducted at our institution [10], (including 1134 cases of proximal humeral nailing with the same implant), the overall complication rate was lower by 12.6%. In contrast to this study, it must be stated that no exclusions regarding follow-up data took place, which may have left several complications unreported.

The debate about the appropriate indication for intramedullary nailing is as old as the operative concept itself. One of the most significant benefits of intramedullary nailing in the elderly is the less invasive approach which renders an exact anatomical reduction much more difficult. Following the AO’s guiding principles 1990, Gerber et al. [21] did not consider anatomical repositioning mandatory if the articular surface is not involved and the reduction restores length and correct configuration in all three planes. This might not be utterly transferable in the case of proximal humeral fractures. Stedtfeld and Mittlmeier [7,8], pointed out the importance of reducing the head and consecutively “closing the curtain” by lowering the tubercles with the “rope over bitt technique.” This was later similarly made clear by Gerber et al. via remonstrating the anatomical reduction as a crucial step in the operative treatment [6].

The collected data showed an anatomical reduction to be highly protective concerning later revisions, implant failures, and OHN, therefore being in line with earlier findings.

Schnetzke et al. [22] have also pointed out that non-anatomical reduction is a major risk factor for OHN, calculating a relative risk of 4.5. The effect presented even more significant in this study with an RR of 8.6.

In having detrimental effects on the functional outcome, humeral head necrosis is one of the most feared and widespread adverse events after surgery of PHFs [23,24]. Hertel et al. [15] pointed out the importance of the medial hinge (medial periosteal area) as a critical structure for restoration of the head perfusion as well as for biomechanical stability [25].

In line with previous findings, the postoperative integrity of the medial hinge showed it to be a significant risk factor for the development of OHN in this study. This confirms earlier findings [15,26,27], underlining the importance of restoring the medial support during anatomical reduction.

Being a central structure concerning the perfusion of the proximal humerus, the periosteum is primarily supplied by the posterior circumflex humeral artery and is closely attached with the entheses of the rotator cuff to the bone. It, therefore, forms a collar-like band around the neck of the proximal humerus [27].

Moreover first described by Hertel et al. [15], an humeral head extension of less than 8 mm is considered a risk factor for ischemia of the humeral head. Therefore, the demonstrated results approve previous findings regarding the used antegrade nailing system [24,28].

Many biomechanical studies have also pointed out the importance of the calcar region (anterior and posterior) for the stability of the proximal humerus [29]. Mehta et al. demonstrated a great loss of fixation strength in a cadaver study if the plate fixation did not include the calcar [30]. Katthagen et al. stressed further in another cadaver study that medial bone block augmentation significantly increased stability, while a second calcar screw did not show a substantial increase in fixation strength [31].

In recent times, intramedullary nails with the improvement of a specific calcar screw have been developed [31,32,33]. Rothstock et al. [34] demonstrated in a biomechanical study the superior fixation strength of the MultiLoc (DePuy Synthes) nail vs. an older version of the Targon PH nail.

Wanzl et al. also showed the biomechanical importance of a calcar screw in the MultiLoc nail and the biomechanical advantage of longer head screws [12].

Having analyzed the new Targon PH^+^ nail, which itself is not equipped with a classical calcar screw but ascending screw design, we found an association between the distance of the distal head screw to the calcar region and a lower tendency of the nail to cut out (*p* = 0.005). This emphasizes the importance of the fixation of an implant in this anatomical region by choosing an adequate screw length.

The PHF is being considered an osteoporotic fracture primarily in the elderly. Therefore, osteoporosis could be considered a risk factor for later complications and adverse events. The demonstrated low values of local bone quality could be a risk factor for the fracture itself. As expected, low values in all three parameters were closely connected with the pre-existing diagnosis of osteoporosis, confirming previous findings [35,36]. However, an association between lowered bone quality and later fracture complexity, complications, revisions, and implant failure could not be found.

### Limitations

This study has several limitations that must be considered. First, the study was conducted retrospectively, with inherent limitations.

The selection bias is considerable difficulty in this study due to a significant number of patients dropping out during the inclusion due to insufficient follow-up data or death. Therefore, the reported complication and implant failure rates are higher than comparable collectives.

Data could also be skewed due to the lower number of four-part fractures included. Furthermore, this study does not report functional outcome parameters and only reports outcomes of a single center with a longtime experience in intramedullary nailing. It has to be noted that the study was also conducted at a single research center. While highly experienced, the limited number of surgeons and the nail used in this specific implant study could have led to a biased result.

## 5. Conclusions

Four-part fractures in an elderly population show considerable complication, revision, and implant-failure rates. Radiologic risk factors for adverse events should be carefully evaluated. Anatomical reduction and fixation near the calcar seem to be crucial for the successful antegrade nailing of complex fractures. To reduce the risk for Humeral head necrosis, reconstruction of the medial hinge and the metaphyseal head extension should be considered.

## Figures and Tables

**Figure 1 jcm-11-02523-f001:**
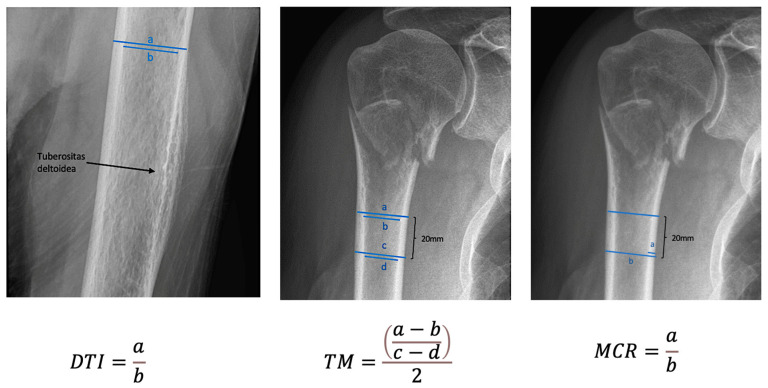
Illustration of the measurement and calculation of the deltoid tuberosity index, Tingart measurement, and the medial cortical ratio.

**Figure 2 jcm-11-02523-f002:**
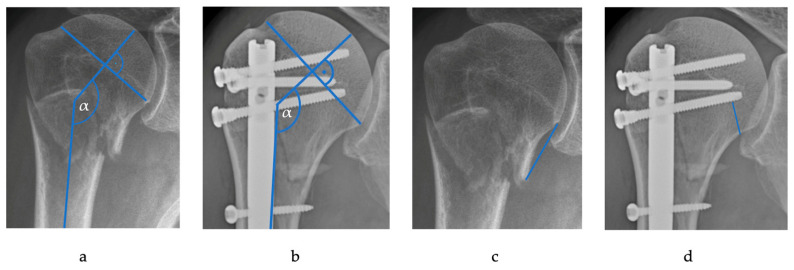
Illustration of measurement of the neck-shaft angle = α (**a**); preoperative and (**b**); postoperative), the metaphyseal head extension (**c**), and measurement of the distance of the distal head screw to the calcar (**d**).

**Table 1 jcm-11-02523-t001:** Patient characteristics: BMI = body mass index, DTI = deltoid tuberosity index, TM = Tingart measurement, MCR = medial cortical ratio, MMHE = medial metaphyseal head extension.

	Mean	Range	±SD
Age in years	75.5	63–94	8.2
BMI	25.8	14.7–46.2	5.2
DTI	1.4	1.2–1.9	0.1
TM	6.2	3.1–13.5	1.6
MCR	0.2	0.1–0.3	0.03
MMHE	17.9	5.0–57.9	11.5
	** *n* **	**%**	
**Gender:**			
Female	83	63.8	
Male	47	35.2	
**Side:**			
Left	69	53.1	
Right	62	46.8	
**Fracture parts:**			
two-part	58	44.6	
three-part	58	44.6	
four-part	15	10.8	
**Neer Classification:**			
III/2-part	58	44.6	
IV/3-part	55	42.2	
IV/4-part	15	11.5	
V/3-part	2	1.5	
VI/3-part	1	0.9	
Pre-existing conditions:	115	88.4%	
Multimorbidity:	48	36.9%	
Osteoporosis:	44	33.8%	
Neurologic diseases:	37	28.5%	
Cardiac disease:	32	24.6%	
Pulmonary disease:	32	24.6%	
Nicotine abuse:	10	7.8%	
Alcohol abuse:	17	13.1%	
Drug abuse:	4	3.1%	

**Table 2 jcm-11-02523-t002:** Implant-specific complication, revision surgery, and implant failures in the entire study population; TSA = total shoulder arthroplasty.

Implant Specific Complication	*n*	% Relative to Study Group	2nd Surgery	Implant Failure *n*	% Relative to Complication
Implant loosening (incl. screw loosening)	7	5.4	1 Removal distal locking screw 6 Re-osteosynthesis	6	85.7%
Cranial nail migration *	6	4.6	1 Revision	0	-
Osteonecrosis of the humeral head	9	6.9	4 Secondary TSAs 3 Removals 1 Revision	4	44.4%
Peri-implant fracture	3	2.3	2 Removals and Reimplantation long nail 1 Removal + plate osteosynthesis	3	100%
Cut out	6	4.6	2 Secondary TSA 3 Removals	4	66.6%
Loss of reposition	6	4.6	3 Secondary TSAs 1 Removal and Reimplantation long nail	3	50%
Nonunion	2	1.5	-	-	-

* Without cutting out of the nail.

**Table 3 jcm-11-02523-t003:** General complications, revision surgeries, and implant failures; TSA = total shoulder arthroplasty.

General Complications	*n*	%	Revision Surgery	Implant Failure *n*	%
Infection	2	1.5	1 Removal and tertiary TSA 1 Removal	2	100%
Joint stiffness	1	0.8	1 Arthroscopic arthrolysis	0	-
Arthralgia	3	2.3	1 revision of screw 1 Removal of the nail	2	66.7%
Hematoma	1	0.8	1 Revision	0	-

**Table 4 jcm-11-02523-t004:** Incidence of complications, revisions, and implant failures in respect to two-, three-, and four-part fractures; * *p*-value of the χ_2_ = Pearson’s chi-square comparing the difference of the three subgroups.

	Complications	Revision	Implant Failure
Two-part (*n* = 58)	18 (31%)	14 (24%)	7 (12%)
Three-part (*n* = 58)	18 (31%)	11 (18%)	10 (17%)
Four-part (*n* = 14)	9 (64%)	9 (64%)	7 (50%)
All (*n* = 130)	45 (35%)	34 (26%)	24 (18%)
*p*-value *	*p* = 0.078	*p* = 0.005	*p* = 0.008

**Table 5 jcm-11-02523-t005:** Comparison of subgroups anatomical vs. non-anatomical reduction regarding complication, revision, and implant failure; AR = anatomical reposition; NAR = non-anatomical reposition; OR = odds ratio, RR = relative risk, OHN= osteonecrosis of the humeral head.

	Complications	Revisions	Implant Failures	Loss of Osteosynthetic Stability *	OHN
AR (*n* = 66)	10 (15%)	7 (10.9%)	1 (1.5%)	3 (4.5%)	1 (1.5%)
NAR (*n* = 64)	35 (54%)	27 (42%)	23 (35%)	17 (26.6%)	8 (11.9%)
AR vs. NAR: **	*p* < 0.000	N/A	N/A	N/A	N/A
AR vs. NAR: #	*p* < 0.0001	*p* < 0.0001	*p* = 0.0001	*p* < 0.0001	*p* = 0.016
OR in NAR-group	7.0	6.5	56.0	9.0	14.0
RR in NAR-group	3.9	4.2	25.1	6.5	8.9

* incl. implant loosening, cut out, loss of reposition and cranial migration of the nail; ** χ_2_ = Pearson’s chi-square test; # Fisher exact test’s *p*-value.

**Table 6 jcm-11-02523-t006:** Incidence of osteonecrosis of the humeral head (OHN) in cases of preoperative disruption of the medial hinge and of no operative reconstruction; IH = intact hinge, DH = disrupted hinge, OR = odds ratio, RR = relative risk, RD = risk reduction.

	Preoperative	Postoperative
**Medial hinge:**		
Intact hinge; *n* (%)	21 (16.2%)	94 (72.3%)
Disrupted hinge; *n* (%)	109 (83.8%)	36 (27.5%)
**Osteonecrosis of the humeral head**		
IH; *n* (%) vs. DH; *n* (%):	9 (9%) vs. 0 (0%)	7 (19.4%) vs. 2 (2.1%)
Fisher exact test’s *p*-value *	0.335	0.002
**Risk for OHN if disruption of the medial hinge:**		
OR	N/A	11.4
RR	N/A	17.6
RD	9%	17.3%

* Comparing OHNs’ distribution of IH and DH versus the rest of the population.

## Data Availability

Not applicable.

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
