# Peer review of "Specific Radiologic Risk Factors for Implant Failure and Osteonecrosis of the Humeral Head after Interlocking Nailing with the Targon PH+ of Proximal Humeral Fractures in a Middle to Old Population"

_jcm, 2022, doi:10.3390/jcm11092523_

Round 1

Reviewer 1 Report

The topic is interesting and the study well designed. The paper is well written although confusing in some parts.

Please provide details of the nail (company,...) at first citation.

Inclusion criteria: age >18 years...it is not geriatric population

How were complications evaluated (classification?)?At which follow up?

Surgical technique should be shortened.

Figure 1 and 2: please show measures on both pre-op and post-op x-rays

Table 1: Neer type not clear

Table 5: not clear

Many grammar errors, should be checked by an english native speaker

Author Response

Dear Editor,

Dear reviewers,

Thank you very much for the effort put in the improvement of our manuscript now entitled:

“Specific radiologic risk factors for implant failure and osteonecrosis of the humeral head after interlocking nailing with the Targon PH+ of proximal humeral fractures in an elderly population”.

We have tried our very best to implement your valuable corrections and proposals.

All changes are marked in the manuscript and detailed replies to reviewers’ comments have been submitted in a separate document.

We hope that the manuscript has improved according to the high standards of the Journal of Clinical Medicine and is now suitable for publication.

Sincerely yours,

Dr. Maximilian Willauschus

Reviewer 2 Report

The manuscript focus on an interesting topic but something in the materials should be improved:

  • it is not a multicentric study and risk analysis for implant failure undergoes the bias due to a single center study.
  • inclusion in criteria are not so strict (only mean age is reported, comorbilities and osteoporosis are not accurately mentioned
  • Fracture pattern is too wide. A most accurate selection on specific pattern should be includerlo.
  • A single type of nail has been studied, with an important surgeon- or implant- dependent bias.

Accoding to these observation, it seems that the results are due to specific implant problems.

Author Response

(The authors gave the same response as above.)

Round 2

Reviewer 1 Report

The Authors made good efforts in the attempt to ameliorate their paper.

However, many points should still be modified.

As inclusion age is >63 years, i would modify title in "middle to old population"

Please provide reasons for removing one of the Authors

Complications: how were assessed?which classification used?

Neer classification: please specify differences between type IV V and VI

Author Response

Dear reviewer, 

on behalf of all authors, we thank you very much for the endeavor put into our manuscript.

We tried to adapt the manuscript according to your suggestions. As in round 1 you'll find the changes made in a separate document.  

Best Regards 

Maximilian Willauschus

Reviewer 2 Report

The manuscript has been improved.

I appreciated authors efforts. 

Author Response

Dear reviewer, 

on behalf of all authors, we thank you very much for the endeavor put into our manuscript. 

M. Willauschus
